# Ethnobotanical Survey of Plants Used by Subsistence Farmers in Mitigating Cabbage and Spinach Diseases in OR Tambo Municipality, South Africa

**DOI:** 10.3390/plants11233215

**Published:** 2022-11-24

**Authors:** James Lwambi Mwinga, Wilfred Otang-Mbeng, Bongani Petros Kubheka, Adeyemi Oladapo Aremu

**Affiliations:** 1Indigenous Knowledge Systems (IKS) Centre, Faculty of Natural and Agricultural Sciences, North-West University, Private Bag X2046, Mmabatho 2790, North-West, South Africa; 2School of Biology and Environmental Sciences, Faculty of Agriculture and Natural Sciences, University of Mpumalanga, Private Bag X11283, Mbombela 1200, Mpumalanga Province, South Africa; 3Dohne Agricultural Development Institute, Private Bag X15, Stutterheim 4930, Eastern Cape, South Africa; 4Discipline of Plant Pathology, School of Agricultural, Earth and Environmental Sciences, University of KwaZulu-Natal, Pietermaritzburg 3209, KwaZulu-Natal, South Africa; 5School of Life Sciences, College of Agriculture, Engineering and Science, University of KwaZulu-Natal (Westville Campus), Private Bag X54001, Durban 4000, KwaZulu-Natal, South Africa

**Keywords:** crop diseases, ethnobotany, food security, phytopathogens, medicinal plants, UN SDGs

## Abstract

Annually, significant crop losses are reported due to diseases caused by phytopathogens. Most subsistence farmers cannot afford the high cost of chemical treatments thereby resulting in the increasing dependence on plant extracts to manage crop diseases. In this study, we documented plants used for the management of cabbage and spinach diseases in OR Tambo Municipality, Eastern Cape Province. An ethnobotanical survey using semi-structured questionnaires was used to document plants and plant parts used by the subsistence farmers in managing cabbage and spinach diseases. Semi-structured questionnaires were administered to 41 consenting subsistence farmers from November to December in 2021, using snowball sampling. The collected data were subjected to descriptive statistical and ethnobotanical analyses. A total of 17 plants belonging to 10 families were identified by the participants as being used in mitigating cabbage and spinach diseases. *Tulbaghia violacea*, *Aloe ferox*, and *Capsicum annuum* had the highest use value of 0.32 each, whereas *Tulbaghia violacea* had the highest relative frequency of citation of 0.39. This current study revealed the importance of plants in managing crop diseases in local communities. It provides baseline data for future pharmacological evaluations in authenticating the efficacies of the identified plants in managing crop diseases.

## 1. Introduction

As one of the United Nations Sustainable Development Goals (UN SDGs), zero hunger (goal 2) is linked to increased agricultural productivity. In the sub-Saharan countries, most people heavily rely on farming as a means of food provision and well-being [1,2]. Over 800 million people living in developing countries are faced with the challenge of food security. In recent years, there has been a global increase in the number of people suffering from hunger, with climate extremes and variability as major drivers negatively affecting food security [3]. Food commodity losses are annually reported worldwide due to crop diseases caused by pathogens. About 10–30% crop loss is caused by microbes and pests [4,5], with microbial pathogens causing the most common soil-borne diseases [6]. Generally, chemical treatments have been used to control crop diseases. However, these treatments have many setbacks, such as negative impacts on plant and soil ecology, natural environment, and public health [7,8,9,10]. The existing crop protection paradigms that rely on synthetic conventional treatments have had minimal impact on the productivity of many poor subsistence farmers in sub-Saharan countries who constitute a major segment of agriculture. With these challenges affecting crop production, there is a need to explore eco-friendly and sustainable methods for crop disease management. The use of plant extracts against crop diseases has gained momentum due to their eco-friendly nature, availability, and biodegradability [11,12,13,14,15]. For instance, various communities in Nigeria use extracts made from *Azadirachta indica*, *Nicotiana tabacum*, *Alstonia boonei*, *Capsicum frutescens*, *Annona squamosa*, *Garcina kola*, *Justicia adhatoda*, *Lantana camara*, *Eucalyptus camaldulensis*, *Moringa oleifera*, *Ocimum* spp., and *Zingiber officinale* to control maize weevil [16,17,18,19]. In the Eastern Cape, some communities use extracts made from *Solanum umtuma, Solanum giganteum*, *Nicotiana tabacum*, *Nicotiana glauca*, *Capsicum annuum*, *Biden pilosa*, *Aloe ferox*, *Aloe vera*, *Allium sativum*, *Allium cepa*, *Tulbaghia violacea,* and *Aloe maculata* to control pests in cabbage [20,21].

In South Africa, Eastern Cape Province holds the third largest share of the country’s commercial agricultural land (37.1%) [22]. The crops mostly cultivated include lemons, watermelons, maize, dry beans, tomatoes, sunflower, cabbage, onion, spinach, guava, carrots, pumpkins, potatoes, pineapples, peaches, oranges, apricots, bananas, and avocado [23]. In Eastern Cape Province, cabbage and spinach are among the most cultivated crops and are regarded as staple food crops [24,25,26,27]. They are inexpensive and rich in nutrients, playing an important role in human diet and health. For instance, spinach has relatively high levels of nutrient content, vitamins A and C, and minerals [26]. These crops can be affected by diseases due to their susceptibility to microbes and pests, as well as climate extremes and variability [27,28]. Some of the diseases affecting the crops include powdery mildew, fusarium wilt, gummy stem blight/black rot, and phytophthora crown and root rot [29].

Plants have been reported to exert antimicrobial effects against phytopathogens [30,31,32]; hence, their utilisation remains significant in the management of crop diseases. However, ethnobotanical studies providing information on the utilisation of plants on crops for disease management against microbes are currently limited in the Eastern Cape Province. Crop diseases caused by microbes are not well-studied, but still cause significant crop losses [33]. As a result, this study focused on the utilisation of plants by subsistence farmers for the management of microbial-related crop diseases in OR Tambo Municipality, Eastern Cape Province, South Africa. The increasing awareness of the need for food security and safety, especially with attaining of relevant UN SDGs (such as zero hunger and good health and well-being), serves as motivation and rationale for this study.

## 2. Results and Discussion

### 2.1. Socio-Demographic Characteristics of Participants

The socio-demographic characteristics of the participants that were collected include location, age, gender, marital status, race, and educational level (Table 1). The 41 participants involved in the study were from four local municipalities; Mhlontlo (12), King Sabata Dalindyebo (26), Nyandeni (1), and Port St. Johns (2). The interviewed participants were Africans. The study indicated that men were more knowledgeable than women, and this could be because men are more engaged in farming in the study area compared with their female counterparts [34]. Indigenous knowledge is possessed among farmers at age groups between 30–44 and 45–59, whereas limited indigenous knowledge exists in the younger generations. These differences in indigenous knowledge can be attributed to changes in lifestyle, industrialization, and urbanisation [35,36].

### 2.2. Sources of Indigenous Knowledge of the Ethnobotanical Information

A crop disease is generally referred to as “izifo zezityalo” in IsiXhosa. Generally, the participants were aware of the microbial-related diseases affecting cabbage and spinach in their production farms. Some farmers tended not to control the diseases because of the high costs of chemical treatments; hence, some crops could die. A few were practising indigenous disease control methods, predominantly the use of medicinal plants, which is consistent with earlier studies by Mlanjeni [37]. The author indicated that rural farmers in the Eastern Cape Province of South Africa used indigenous ethnobiological methods to control stalk borers in maize. The majority of the participants had extensive (over 10 years) experience in farming, and most of them gained knowledge on the use of plants in controlling crop diseases through training from more experienced experts by attended workshops (47%), whereas some gained the knowledge from parents and grandparents (42%) (Figure 1). This finding is supported by the study in Mpumalanga Province of South Africa by Seile et al. [38]. The authors revealed that the indigenous knowledge is gained from grandparents and through training (“uku-thwasa”) with experienced experts and trainers (“Gobela”). In Limpopo Province of South Africa [39], and Namibia [40], traditional health practitioners also learn indigenous knowledge on the use of medicinal plants through training. Indigenous knowledge (including ethnobotanical information) is well-treasured and often shared between family members and communities for its preservation. In Eastern Cape Province, indigenous knowledge on the use of plants to control crop pests and diseases was available, but it was being eroded due to lack of documentation [41].

### 2.3. Diversity of Plant Species Used for the Management of Cabbage and Spinach Diseases in OR Tambo Municipality

A total of 17 plants belonging to 10 families were reported by the participants (Table 2). Asteraceae (30%) and Asphodelaceae (30%) were the most represented families, followed by Solanaceae (20%), and Amaryllidaceae (20%). Asteraceae comprises an interesting group of plants that are viewed as natural alternatives for crop protection. The majority of the members in the family have allelopathic properties, and hence may offer crop protection [42]. Most of the plant species reported were herbs (59%), which is similar to a study by Skenjana and Poswal [20], where the majority of the plants reported to be used for pest control in cabbage were herbs. The rest of the plant species were shrubs (35%) and trees (6%). In Eastern Cape, the use of plant extracts still plays a bigger role in crop protection, likely due to their accessibility and availability. However, most of the ethnobotanical studies conducted have focused on medicinal plants that are used to control pests, insects, and nematodes [20,43]. Currently, only limited data exist on plants used in controlling crop diseases caused by microbes. The documented data will help to better understand the relationship that exists between humans, plants, and biodiversity [44]. Besides crop protection, these plants have been reported to provide basic health care needs for humans and animals [45,46,47].

*Tulbaghia violacea* Harv., *Aloe ferox* Mill., and *Capsicum annuum* L. had the highest use values of 0.32 each, whereas *Tulbaghia violacea* had the highest relative frequency of citation value of 0.39 (Table 2). *Tulbaghia violacea* (leaves), *Aloe ferox* (leaves), *Allium cepa* (leaves and bulb), and *Capsicum annuum* (pods) have also been reported to have the potential to control pests in cabbage [20]. *Aloe ferox* (leaves), *Allium cepa* (seeds), and *Capsicum annuum* (fruits) are known to control pathogens and pests associated with potato [48,49,50]. Besides controlling crop pathogens, *Capiscum annuum* and *Nicotiana tabacum* have been reported to control pests on field crops, whereas *Tagetes minuta* is effective for controlling pests of stored crops [51]. The study by Skenjana and Poswal [20] alluded to the pesticidal activity of *Nicotiana tabacum*, *Capsicum annuum*, *Tagetes minuta*, *Artemisia afra*, *Tulbaghia violacea*, *Allium cepa*, and *Aloe ferox* for the protection of cabbage, which inevitably contributes to food security. According to the participants, these medicinal plants were locally available, cost effective, and environmentally friendly. They also indicated that harvested cabbage and spinach remain fresh and fit for consumption longer when medicinal plants are used, rather than synthetic chemical treatments, in controlling crop diseases.

**Table 2 plants-11-03215-t002:** Inventory of plants used by subsistence farmers in mitigating diseases of cabbage and spinach in OR Tambo Municipality, South Africa. Verification of the botanical names was done using the ‘World Flora Online’ [52].

Scientific Name (Voucher Number)	Family	Local Name	Habit	Part Used	Mode of Preparation and Application	* Targeted Disease(s)	UV	RFC
*Acacia karoo* Hayne (LM 01)	Fabaceae	Umnga	Shrub	Leaves	Leaves are macerated and sprayed on the crop.	C7	0.02	0.02
*Agave americana* L. (LM 02)	Asparagaceae	Ikhamanga	Herb	Whole plant	Whole plant is macerated and sprayed on the crop.	S3	0.02	0.02
*Aloe ferox* Mill. (LM 03)	Asphodelaceae	Ikhala	Herb	Leaves	Leaves are either macerated or its decoction sprayed on the crop, applied in the soil, or at spot of disease.	S1, S2, S3, S4, S5, C1, C2, C3, C4, C5, C6, C7, C8, C9	0.32	0.34
*Allium cepa* L. (LM 04)	Amaryllidaceae	Onions	Herb	Bulb	Decoction of the bulb is sprayed on the crop.	S1, S3, C1, C2, C3, C4, C5, C6, C7	0.22	0.34
*Artemisia afra* Jacq. (LM 05)	Asteraceae	Umhlonyana	Shrub	Whole plant	Whole plant is either macerated or its decoction sprayed on the crop, or applied at spot of disease.	C2, C5, S5	0.07	0.07
*Bulbine frutescens* Willd. (LM 06)	Asphodelaceae	Bulbine	Shrub	Whole plant	Decoction of the whole plant is applied on the spot of disease.	C8	0.02	0.02
*Capsicum annuum* L. (LM 07)	Solanaceae	Ikhanakhana	Shrub	Fruit	Fruit is either macerated or its decoction sprayed on the crop, or applied at spot of disease.	S1, S2, S3, S4, S5, C1, C2, C3, C4, C5, C6, C7, C8, C9	0.32	0.32
*Datura stramonium* Thunb. (LM 08)	Solanaceae	Vumbangwe	Herb	Whole plant	Whole plant is either macerated or its decoction sprayed on the crop.	S1, S2, S3, S4, S5, C1, C2, C3, C4, C5, C6, C7, C8, C9	0.32	0.20
*Eucalyptus camaldulensis* Dehnh. (LM 09)	Myrtaceae	Gumtree	Tree	Leaves	Leaves are either macerated or its decoction sprayed on the crop.	C2, C5, C7, S4, S6	0.12	0.05
*Exomis microphylla* (Thunb.) Aellen (LM 10)	Amaranthaceae	Mvenyathi	Shrub	Roots	A paste made from the roots is applied at the spot of disease.	C5	0.02	0.02
*Helianthus annuus* L. (LM11)	Asteraceae	Sunflower	Herb	Roots	Roots are macerated and sprayed on the crop.	C1	0.02	0.02
*Kniphofia uvaria* (L.) Oken (LM 12)	Asphodeladeace	Hot poker	Herb	Leaves, stem	Decoction of the leaves or stem is applied on the spot of disease.	C9, S3	0.05	0.05
*Nicotiana tabacum* Vell. (LM 13)	Solanaceae	Icuba lesixhosa	Herb	Leaves	Leaves are either macerated or its decoction sprayed on the crop.	S1, S2, S3, S4, S5, C1, C2, C3, C4, C5, C6, C7, C8, C9	0.32	0.29
*Ptaeroxylon obliquum* (Thunb.) Radlk. (LM 14)	Rutaceae	Umthathi	Shrub	Leaves	Decoction of the leaves is sprayed on the crop.	S3	0.02	0.02
*Tagetes minuta* L. (LM 15)	Asteraceae	Nukanuka/Isangu	Herb	Whole plant	Whole plant is either macerated or its decoction sprayed on the crop. Its paste applied at the affected area of the plant.	S1, S2, S3, S4, S5, C1, C2, C3, C4, C5, C6, C7, C8, C9	0.32	0.24
*Tulbaghia violacea* Harv. (LM 16)	Amaryllidaceae	Isivumbampunzi	Herb	Whole plant	Whole plant is either macerated or decoction sprayed on the crop, drenched in the soil, or its paste can also be applied at the affected area of the plant	S1, S2, S3, S4, S5, C1, C2, C3, C4, C5, C6, C7, C8, C9	0.32	0.39
*Zantedeschia aethiopica* Spreng. (LM 17)	Araceae	Nyibiba	Herb	Whole plant	Whole plant decoction is drenched in the soil or applied at the affected area of the plant.	C8, S4	0.05	0.05

* Targeted disease(s)—cabbage diseases: C1 = sclerotina, C2 = ring spot, C3 = fusarium wilt, C4 = downy mildew, C5 = powdery mildew, C6 = alternaria leaf spot, C7 = bacterial spot, C8 = root rot, C9 = black rot. Spinach diseases: S1 = anthracnose, S2 = Cladosporium leaf spot, S3 = damping off and root rot, S4 = downy mildew, S5 = Stemphylium leaf spot. UV = use value. RFC = relative frequency of citation.

### 2.4. Mode of Utilisation of the Reported Plants

In the current study, leaves (46%) were the most frequently used plant part, followed by whole plant (23%), whereas flowers and bark were the least used plant parts, each with a frequency of 0.98% (Figure 2). Leaves were frequently used and this could be due to their availability, abundance, and ease to harvest and work with compared with other parts of the plant [53]. Generally, leaves have a high quantity of phytochemicals [54], and this could be attributed to the process of photosynthesis, which produces secondary compounds effective against most diseases and pests [55,56,57].

The participants made use of different procedures and formulations to prepare the plant extracts. The different procedures identified by farmers were categorised using three extraction techniques, namely decoction (boiling in water), maceration (soaking in water), and pasting. To make a paste, the plant part was first crushed, then mixed with water. Decoction (50.49%) was the most used technique, followed by maceration (46.6%). Water was the only solvent used for extraction. Herbalists mostly formulate medicinal plant recipes using water extracts, which is seen as convenient and effective [58,59].

The plant extracts were applied by spraying on the crop, injecting in soil, or spot treatment at point of disease. Spraying was the most used method of application (86.41%), and a few participants used both soil injection and spot treatment. Spraying was deemed to be a more direct and efficient mode of application with potential for reliable and holistic results. According to the farmers, the frequency of application of the extracts on the crops depended on the availability of plant materials and occurrence of disease. Some farmers applied the extracts on a regular basis as preventative measures against crop diseases. Most participants applied the extracts at sight of disease (37.86%), followed by once in a week (26.21), and twice a week (17.48%).

### 2.5. Plant Combination Remedies

The practice of using a combination of plants or addition of non-plant materials in mitigating crop diseases was evident in this current study. Some participants indicated the utilisation of plant combination remedies in preventing and mitigating crop diseases in cabbage and spinach. The remedies are reported by the participants to be cost-effective and effective in managing crop diseases.

Six participants mentioned the use of three plant combination remedies. In the first remedy, a bulb of onion (*Allium cepa*) and garlic (*Allium sativum*) are chopped and mixed with 1 tablespoon of crushed red pepper (*Capsicum annuum*). The mixture is boiled gently for 20 min in 1.5 L of water. The resulting decoction is filtered through a piece of cloth into a bottle, sealed and stored for 6 weeks. Before use, 750 mL of water is added to the recipe, then sprayed on the crop or flicked on the crop with a brush. In the second remedy, three onions are chopped and soaked for two days in 2 teaspoons of paraffin. Thereafter, 1.5 L of water is added and shaken to thoroughly mix, which is followed by the addition of 1 tablespoon of dishwashing liquid to the mixture, sealed, and allowed to stand for one day. Before use, 2 teaspoons of the mixture are added to 1 L of water and sprayed on the crop or flicked on the crop with a brush. The last remedy involves placing a mixture of chopped chillies (fruits), tobacco (leaves), onions (bulb), and wild garlic (whole plant) in a sack cloth, then suspended in a bucket of water. The contents suspended in the water are gently boiled for 20 min. After cooling, the contents in the sack cloth are emptied in the water. Paraffin and dishwashing liquid are added to the water, and the bucket is sealed and stored for one week. The mixture is sprayed on crops for prevention and treating of crop diseases.

The use of these aforementioned plant combination remedies by the subsistence farmers in OR Tambo to make extracts for managing crops diseases in this current study was consistent with farmers from other parts of Eastern Cape [20], other parts of South Africa [60], and Africa [61,62,63,64]. The use of plant combination remedies alludes to the multi-purpose characteristics and holistic medicinal effects of the extracts [47,65], capable of mitigating many types of diseases and pests.

### 2.6. Availability and Conservation Status of the Reported Plants

A total of 10 plants were reported to be harvested as a whole, which included harvesting the roots and bulb. Harvesting the whole plant has an impact on its future existence. Leaves were observed to be the most harvested plant part; this is relatively more sustainable as plants can easily regrow new leaves. Based on the SANBI Red Data list, most of the reported plants have less threat on their existence as they are termed to be of ‘least concern’ [66]. Only seven of the reported plants were indigenous to Eastern Cape, South Africa (Table 3). However, their continuous exploitation without sustainable means of conserving their regeneration could pose a threat to their future existence.

Some participants observed scarcity of some plants and showed concern for the dwindling of their existence. To address this concern, some of the participants have resorted to cultivating these plants in their backyards. The cultivated plants include *Allium cepa*, *Capsicum annuum*, and *Nicotiana tabacum*. They also discouraged harvesting of the plants at certain seasons, mostly during the winter. In addition, they discouraged harvesting the whole plant, rather than parts of the plant, with an aim to promote future existence of the plant.

## 3. Materials and Methods

### 3.1. Study Area

The research was conducted in OR Tambo Municipality (Figure 3), Eastern Cape Province, South Africa, which lies within latitude −31.4674 and longitude 29.05247. It has five local municipalities, namely Nyandeni, Mhlontlo, Port St Johns, King Sabata Dalinyebo, and Ingquza. The vast majority (94%) of the 1,364,943 people in OR Tambo Municipality speak isiXhosa. The Eastern Cape is located along the coastal line in the south-eastern part of South Africa. It has an area of about 168 966 km^2^ and is considered the second largest province in South Africa [67]. As the second largest province, it accounts for 9.7% of South Africa’s agricultural production [68]. It has the third largest share of the country’s commercial agricultural land (37.1%), with plentiful grassland and good rainfalls [22]. The edapho-climatic conditions of the Eastern Cape support high plant diversity [69]. For this study, Qumbu, Blackhill, Tsolo bridge, Godzi, Gongathe, Marhambeni, Ncise, Payne, Nongcwena, Upper Tabase, Sitholeni, Mqobiso, Kotishini, Upper Ntafufu, and Ntimbini ward 7 locations were randomly sampled using the probability method. With the non-probability method, a snowball technique was used to select the subsistence farmers to be interviewed in the study area [70].

### 3.2. Ethnobotanical Data Collection

An ethnobotanical survey was conducted in OR Tambo between November and December 2021. A team of research assistants who were also residents familiar with the local language helped with the identification of knowledge holders. All interviews were strictly conducted under COVID-19 rules and regulations, as stipulated by the World Health Organization and the South African government. Semi-structured questionnaires were administered to the knowledge holders to obtain information on plants used to manage microbial diseases in cabbage and spinach. The first section of the questionnaire focused on the socio-demographic information of the participants. The second section captured information on the medicinal plants. To identify the specific diseases that affect cabbage and spinach, the participants were presented with photographs of common diseases affecting cabbage and spinach in Eastern Cape, South Africa (Appendix A). The interviews were supplemented with field trips where plant materials were collected from their natural population around the OR Tambo Municipality in Eastern Cape Province, under the guidance of some of the participants. Preliminary identification of the plants was done based on the scientific works of Dold and Cocks [71] and Van Wyk, Oudtshoorn [72]. Voucher specimens of the plants were prepared and deposited at the S.D. Phalatse Herbarium, North-West University, Mafikeng Campus.

### 3.3. Ethical Consideration

Ethical clearance (NWU-00530-21-A9) was obtained from the Faculty of Natural and Agricultural Sciences Research Ethics Committee (FNASREC). Permission to conduct research in the study area was obtained from the traditional leaders through the traditional leader of the Xorana administration area. The participants were given consent forms, where the aim and objectives of the research was explained to them; they gave their consent prior to being interviewed. It was also explained to them that their identity would be kept confidential, that participating in the research was voluntarily, and that they could withdraw at any time if they felt uncomfortable. The participants also signed non-disclosure agreement forms before commencing the research. The participants were identified using codes.

### 3.4. Data Analyses

The collected demographic data of the participants and ethnobotanical information of the plants were compiled in a database in an Excel workbook, coded, and then categorized according to their main themes [73]. Descriptive statistics were used to analyse the socio-demographic data of the participants, whereas ethnobotanical indices were used to summarize the ethnobotanical information of the plants. The information was analysed using frequency, use value (UV), and relative frequency of citation (RFC).

The percentage of knowledge holders claiming the use of a certain plant was calculated as:(1)$$F\;(\%)=\frac{\mathrm{Np}}{N}\times 100$$
where Np is the number of participants that claim a particular use of a plant and N is the total number of participants.

The use value alludes to the relative significance of the locally known material, hereby referring to the local communities of the OR Tambo. As indicated by Tardío and Pardo-de-Santayana [74], it was calculated as the total number of uses of a plant mentioned by the participants divided by the total number of participants taking part in the survey.

The relative frequency of citation considers the spread of use (number of participants) for each plant, along with its versatility, which is a measure of the diversity of applications of the plant species [74]. The index was calculated as the sum of all participant mentions of each plant divided by the total number of participants taking part in the survey.

## 4. Conclusions

The participants identified 17 plants (10 families) as being used for mitigating microbial-related diseases in cabbage and spinach. The frequency, UV, and RFC values identified *Tulbaghia violacea*, *Aloe ferox*, *Allium cepa*, and *Capsicum annuum* as the most valued medicinally used plants in the management of diseases affecting cabbage and spinach. This is an indication of the importance of these plants in the protection of cabbage and spinach against phytopathogens in Eastern Cape Province. In addition, the associated indigenous knowledge and practices related to the identified plants were recorded. This indigenous knowledge points to great potential for further research in crop protection and food security. Further exploratory surveys on plant use in crop protection against microbes need to be conducted, and the knowledge preserved through documentation. Promoting a sustainable harvesting pattern of these medicinal plants is of paramount importance to ensure their future existence.

## Figures and Tables

**Figure 1 plants-11-03215-f001:**
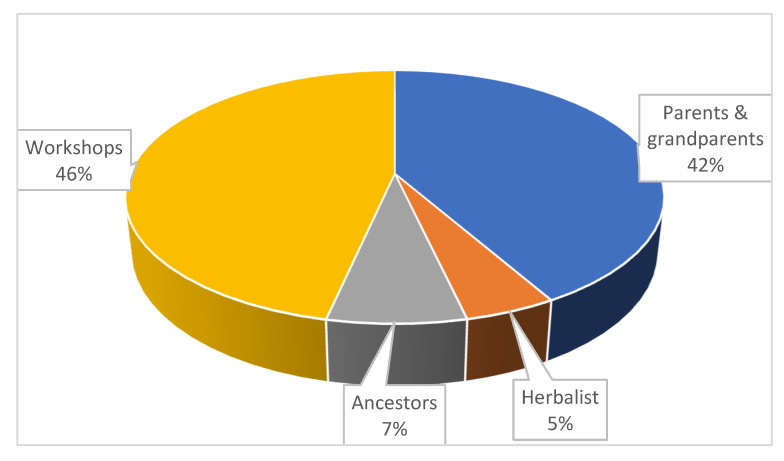
Distribution of the sources of indigenous knowledge among subsistence farmers who use medicinal plants for mitigating cabbage and spinach diseases in OR Tambo Municipality, South Africa, (n = 41).

**Figure 2 plants-11-03215-f002:**
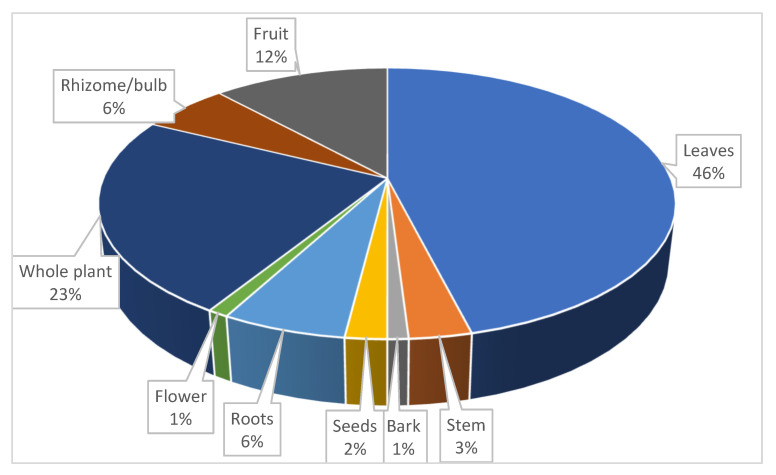
Distribution of plant parts used by the subsistence farmers in mitigating cabbage and spinach diseases in OR Tambo Municipality, South Africa, (n = 103).

**Figure 3 plants-11-03215-f003:**
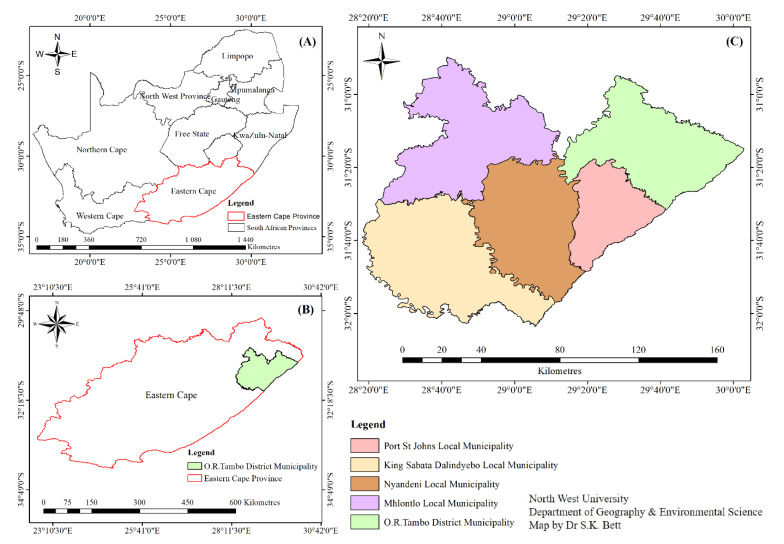
Study areas located in Eastern Cape Province, South Africa: (**A**) South Africa; (**B**) OR Tambo District Municipality; and (**C**) selected local municipalities consisting of Nyandeni, Mhlontlo, Port St Johns, and King Sabata Dalinyebo local municipalities.

**Table 1 plants-11-03215-t001:** Socio-demographic characteristics of the participants who use plants in mitigating cabbage and spinach diseases in OR Tambo Municipality, South Africa, (n = 41).

Parameter		Number of Participants	Frequency (%)
Local Municipality/Location			
Mhlontlo	Qumbu	1	2.44
	Blackhill	1	2.44
	Tsolo bridge	5	12.20
	Godzi	4	9.76
	Gongathe	1	2.44
King Sabata Dalindyebo	Marhambeni	2	4.88
	Ncise	7	17.07
	Payne	1	2.44
	Nongcwena	1	2.44
	Upper Tabase	5	12.20
	Sitholeni	7	17.07
	Mqobiso	3	7.32
Nyandeni	Kotishini	1	2.44
Port St. Johns	Upper Ntafufu	1	2.44
	Ntsimbini ward 7	1	2.44
Age	18–29	7	17.07
	30–44	13	31.71
	45–59	13	31.71
	Above 60	8	19.51
Gender	Male	23	56.10
	Female	18	43.90
Marital Status	Single	15	36.59
	Married	19	46.34
	Separated	3	7.32
	Divorced	1	2.44
	Widowed	3	7.32
Education Level	Informal	13	31.71
	Matric	13	31.71
	Tertiary	15	36.59

**Table 3 plants-11-03215-t003:** Conservation status and availability of reported plant species used by subsistence farmers in mitigating cabbage and spinach diseases in OR Tambo Municipality, South Africa.

Plant	Distribution [66]	Cultivated Part	Status as Reported by Farmer	National Status [66]
*Acacia karoo*	Not available	Leaves	Abundant	Least concern
*Agave americana*	Naturalized exotic	Whole plant	Rare	Not evaluated
*Aloe ferox*	Indigenous	Leaves	Fairly abundant	Least concern
*Allium cepa*	Not available	Bulb	Abundant	Not available
*Artemisia afra*	Indigenous	Whole plant	Rare	Least concern
*Bulbine frutescens*	Indigenous	Whole plant	Rare	Least concern
*Capsicum annuum*	Naturalized exotic	Fruit	Fairly abundant	Not evaluated
*Datura stramonium*	Naturalized exotic	Whole plant	Abundant	Not available
*Eucalyptus camaldulensis*	Naturalized exotic	Leaves	Rare	Not evaluated
*Exomis microphylla*	Naturalized exotic	Roots	Abundant	Least concern
*Helianthus annuus*	Naturalized exotic	Roots	Abundant	Not available
*Kniphofia uvaria*	Indigenous	Leaves, stem	Rare	Least concern
*Nicotiana tabacum*	Naturalized exotic	Leaves	Abundant	Not evaluated
*Ptaeroxylon obliquum*	Indigenous	Leaves	Abundant	Least concern
*Tagetes minuta*	Naturalized exotic	Whole plant	Fairly abundant	Not evaluated
*Tulbaghia violacea*	Indigenous	Whole plant	Fairly abundant	Least concern
*Zantedeschia aethiopica*	Indigenous	Whole plant	Rare	Least concern

## Data Availability

Not applicable.

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
