# Peer review of "Ethnobotanical Survey of Plants Used by Subsistence Farmers in Mitigating Cabbage and Spinach Diseases in OR Tambo Municipality, South Africa"

_plants, 2022, doi:10.3390/plants11233215_

Round 1

Reviewer 1 Report

The manuscript of James Lwambi Mwinga et al entitled " Ethnobotanical survey of plants used by subsistence farmers in mitigating cabbage and spinach diseases in the OR Tambo Municipality, South Africa" is an interesting article which documents the usage of plants by farmers in managing cabbage and spinach diseases. This work shows the importance of plants as eco-friendly and sustainable method for crop disease management. The study is really interesting, while I have no major concern about the overall study design and results, I have few minor comments.

Comments

·         Though, authors have mentioned about the deposition of herbarium specimens in a herbarium. Does is it possible for the authors to provide the digital scan copy of the herbariums as a supplementary material along with the collection site details?

·         In table 2, authors have indicated targeted diseases and have codes as C1-C9 and S1-S5. My question is how authors converted the farmer’s description of microbial disease to scientific names (for example, fusarium wilt). What the farmers described about fusarium wilt, and how the authors confirmed that the farmers are talking about fusarium wilt. Did authors follow any standards for such conversion traditional knowledge to scientific knowledge.

·         Authors could provide more details about the farmers perception of microbial disease, how many types of microbial disease that they aware of? Whether the authors have provided them any new knowledge (Scientific classification of microbes) back to the farmers about the microbial disease.

·         Table 2 can be modified by adding the family name next to the species name separated by a semi-colon. Authors may use the term “habit” rather than using Plant life form in table 2.

·         In table 1, authors may remove the sub-section race, since there are no white and colored people interviewed. Authors may write a line in the methodology that the interviewed farmers were Africans.

·         Authors state that the “shelf-life increases when using the medicinal plants than using synthetic chemicals”, authors can elaborate these aspects. Shelf-life in the sense, for hours or days OR is it related to freshness or the appearance of cabbage and spinach.

Reviewer 2 Report

The manuscript „Ethnobotanical survey of plants used by subsistence farmers in mitigating cabbage and spinach diseases in the OR Tambo Municipality, South Africa“ by Mwinga et al. provides valuable data on the plant species used by subsistence farmers from OR Tambo Municipality, South Africa, against microbial-related diseases of two important crops, cabbage and spinach. The research seems to be well conducted and the manuscript is well prepared and provides the obtained results in a clear and concise manner.

Note that three more species had the use-value of 0.32 other than those stated in the Abstract.

Minor typing errors need to be corrected:

Line 55 – Lantana camara

Line 56 – Ocimum spp.

Line 59 – Bidens pilosa, Aloe vera

Line 145 – Nicotiana tabacum

Table 2 – Datura stramonium, Nicotiana tabacum, Asphodelaceae (see Kniphofia), Cladosporium and Stemphylium should be in italic

Table 3 – Datura stramonium

Check reference 38 (double period).

Other remarks:

Use all caps in the article title, tables (e.g., Number of Participants, Local Municipality, Marital Status…)

Table 1 - Age, Gender etc. could all be put in bold the same as Local Municipality.

Adjust Tables 2 and 3 according to Instructions for Authors. Journal names should be abbreviated.
